# Investigations of Liquid Steel Viscosity and Its Impact as the Initial Parameter on Modeling of the Steel Flow through the Tundish

**DOI:** 10.3390/ma13215025

**Published:** 2020-11-07

**Authors:** Marta Ślęzak, Marek Warzecha

**Affiliations:** 1Department of Ferrous Metallurgy, Faculty of Metals Engineering and Industrial Computer Science, AGH University of Science and Technology, Al. Mickiewicza 30, 30-059 Kraków, Poland; 2Department of Metallurgy and Metal Technology, Faculty of Production Engineering and Materials Technology, Częstochowa University of Technology, Al. Armii Krajowej 19, 42-201 Częstochowa, Poland; marek.warzecha@pcz.pl

**Keywords:** steel, viscosity, tundish, numerical modelling

## Abstract

The paper presents research carried out to experimentally determine the dynamic viscosity of selected iron solutions. A high temperature rheometer with an air bearing was used for the tests, and ANSYS Fluent commercial software was used for numerical simulations. The experimental results obtained are, on average, lower by half than the values of the dynamic viscosity coefficient of liquid steel adopted during fluid flow modeling. Numerical simulations were carried out, taking into account the viscosity standard adopted for most numerical calculations and the average value of the obtained experimental dynamic viscosity of the analyzed iron solutions. Both qualitative and quantitative analysis showed differences in the flow structure of liquid steel in the tundish, in particular in the predicted values and the velocity profile distribution. However, these differences are not significant. In addition, the work analyzed two different rheological models—including one of our own—to describe the dynamic viscosity of liquid steel, so that in the future, the experimental stage could be replaced by calculating the value of the dynamic viscosity coefficient of liquid steel using one equation. The results obtained support the use of the author’s rheological model for the above; however, this model still needs to be refined and extended to a wide range of alloying elements, mainly the extension of the carbon range.

## 1. Introduction

Today, the continuous casting process is the dominant method of steel casting. The basic aggregates of the steel continuous casting machine are tundish and mold. The multitude of functions fulfilled by the tundish is the reason for the undertaking of many studies by scientists and steelmakers from all over the world of science and industry. The positive aspects of the flow of liquid steel through the tundish are widely used in practice. Research undertaken by scientists is still necessary to identify specific parameters of technological processes and to further optimize the process. They can be partly carried out directly on devices operating in industrial conditions, using specific control and measuring devices. The results of such research (called industrial) are of great practical importance, as they enable their direct application in real conditions. Unfortunately, due to the specificity of the processes carried out in metallurgical devices, conducting industrial research is limited in scope and usually concerns individual measurements. In a situation where it is impossible, difficult and/or expensive to carry out relevant tests directly on a working device, model tests using physical [1,2] or numerical [3,4,5] modeling techniques are conducted. Due to the simplifications used, such research also has some limitations, but they have nevertheless become equal research tools for scientists. This applies to both laboratory tests carried out on physical models and numerical tests carried out using computational fluid dynamics (CFD). Simulating the process is cheaper and requires much less input, material or human.

When conducting numerical simulations, one should be aware not only that the mathematical models used are simplified, but also how the simplification affects the reliability of the results obtained using them. When modeling the flow of liquid steel in steel aggregates, including the tundish, parameters such as density and viscosity of liquid steel are often taken from the literature data (the most commonly adopted density of liquid steel is about 7000 kg/m^3^ for the ladle process and 7010–7040 kg/m^3^ for flows in tundish, while the viscosity is about 0.006–0.007 Pas). How big a mistake do we make in this way? The calculation results presented in this article are an attempt to at least partially answer the above questions. Rheological tests were carried out to experimentally determine the viscosity of the tested steel grades. Rheology is interested in the behavior of bodies under the influence of force applied to them. Reverse issues, i.e., the potential response of the system to the given stress, are also of interest to this field of science [6,7]. In rheological terms, we are interested in bodies in various states of aggregation: solid, liquid, solid–liquid [8,9,10].

In this work, rheology has been included in relation to material problems, and the area of interest has become iron solutions and their viscosity. In the literature on the subject, there are numerical data showing that the physical quantity, which is viscosity, is low in the case of iron solutions, similar to water viscosity [11,12]. There are also theories that completely liquid steel is a Newtonian ideal. Thus, it is assumed that the viscosity, i.e., the ratio of shear stress to shear rate [6,7,8,9], depends only on temperature changes, while independent of the force with which we interact with the system. The conclusion is that the viscosity curve of Newton’s perfectly viscous fluid is always a straight line, regardless of changes in the shear rate value, i.e., the force that we give to the system under test [6,7,8].However, the development of today’s technology has also resulted in the development of measuring apparatus that can allow the verification and testing within much wider limits and using different values of measurement parameters. What is more, the sensitivity of the devices can capture the differences, often very small, between individual species of iron solutions that differ in their chemical composition. At present, the subject of rheology of iron alloys has been mainly dealt with in relation to metal flow and the possibility of its formation in liquid states, in relation to the so-called thixoforming [13,14,15]. Often, these studies were accompanied by experiments conducted with light metals, mainly aluminum and magnesium [16,17,18,19,20,21,22]. The authors have previously conducted rheological studies in relation to both liquid and solid–liquid iron solutions, from the point of view of observing the flow of material used in various metallurgical processes [23,24,25].

The authors of this work made an effort to experimentally determine the viscosity of the steel grades they were investigating as part of a larger project. In the next stage, they checked—through model tests—how the introduction of measured values into the numerical model would affect the numerically obtained steel flow forecasts through the tundish, in particular on the flow structure and velocity profile.

## 2. Experimental and Numerical Investigations

Data from real rheological measurements can be used to model the process of metal behavior in the tundish of a continuous steel casting device. In software used for modeling the fluid flow process, there are usually built-in thermodynamic bases that contain the values of basic physical quantities [12]. However, with this approach to the problem and issues, we return to the use of unitary, single data, while the problem can be much more complex.

In rheological terms, the value of dynamic viscosity coefficient is not a single point, but the curve is drawn for different temperatures, for different chemical compositions, using variable values of rheological parameters (shear stress, shear rate) [26]. Only with this look can you fully understand the process and try to reproduce it by computer simulation. In the real process, the system is not affected—most often—by one force of the same size, but we deal with a system of forces and a complex problem and phenomenon. Thanks to modern research equipment and modern software, it is possible to carry out high-temperature, advanced measurements and reproduce the entire complexity of encountered processes. Thanks to the use of experimental and numerical studies, the paper presents what impact some assumptions (simplifications) may have when applied to the initial and boundary conditions of the numerical model of steel flow through the tundish, in particular the liquid steel parameter—which is viscosity—on changing the flow structure in the working space of a tundish.

### 2.1. Calibration Measurements

High-temperature rheological tests were performed on the FRS1600 rheometer equipped with a standard DSR301 head (Figure 1). This device is a prototype machine used to study high-temperature metallurgical systems, and was developed in cooperation with employees of Anton Paar GmbH (Graz, Austria) and the AGH Department of Ferrous Metallurgy (Kraków, Poland). The rheometer is equipped with a resistance tube furnace, enabling the temperature in the sample to be reached to about 1530 °C. In addition, during the measurement, the sample resides in a protective atmosphere of argon, fed into the furnace from below.

The high-temperature rheometer is a very sensitive device (torque accuracy of 0.001 mNm), and the head is equipped with an air bearing. In this rheometer, the torque values are measured by the head and then the software calculates the values of shear stress, viscosity etc. The parameters of the geometry of using measurement system are implemented in the Rheoplus software (Anton Paar GmbH, Graz, Austria ) before starting the experiments.

The control of the furnace is possible from the rheometer software, which allows us to program the experiments with changes of temperature.

Temperature is measured in the furnace (thermocouple in the furnace) and then the temperature in the sample is calculated automatically by the software. In the Rheoplus software, the temperature calibration equation was implemented. The accuracy of temperature measurement is verified during high-temperature measurement of the viscosity pattern.

The apparatus can carry out rotational and oscillation tests as well as material softening tests. The measurement is made in a stationary crucible, in which a sample of the tested material is placed. Inside the crucible, the spindle is immersed in the tested material. The crucible is placed inside the ceramic protective tube, which is part of the heating furnace. The furnace is made of four SiC heating elements heated by electricity. The whole is protected from the outside with insulating material. The temperature inside the furnace is controlled by changing the power supply in the measuring and control system. The rate of heating and maintaining a constant temperature are set in the Rheoplus Rheometer software control panel. The measuring head driven by an electric motor controls the rotational movements of the spindle. The spindle is suspended on a ceramic tube located in the air bearing. To ensure low temperatures, the head is cooled by water and air. The rheometer uses a concentric cylinder system with a rotary internal cylinder. This is called the Searle measuring system [27].

For these tests, a measuring spindle with a perforated side surface, 26.6 mm in diameter, and a crucible with smooth internal walls with an internal diameter of 30 mm were used. Therefore, the measurements were carried out in a normalized system with a narrow shear gap, described in ISO3219 and DIN53019 [27]. The diagram below (Figure 2) presents the measuring range of the device, using a system with an internal cylinder with a diameter of 26.6 mm (for simplicity marked with the number 27) and the symbol CC (concentric cylinder).

The measuring range of the device is very large. Even when using the “large” spindle and conducting research in so-called narrow shear gap, we can test a variety of substances, from water to liquid steel and even liquid glass.

Figure 3 schematically presents the measuring system: internal (bob—Figure 3a) and external (cup—Figure 3b) used. Materials for measuring tools for testing were selected in such a way that the surface of the tool did not react with the tested sample. The tools are made of zirconia-stabilized aluminum oxide (96% Al_2_O_3_ + 1% ZrO_2_).

Before performing the correct rheological measurements, a motor adjustment test and measurement in the air (air check) were performed (Figure 4), to verify the technical condition of the device. During the measurement, the torque on the measuring head was measured along with the change in the yaw angle. The value of the measured torque (blue curve) should be within the error limits of ±0.05 µN (red dashed lines).

The rheometer works correctly and the air bearing is functional if the measured torque is within the error range with the increasing angle of deflection. The resulting chart indicates that the device is in very good condition. Before the measurements were started, the measuring spindle inertia was calibrated, thanks to which the correction was made for a given tool used for measurements. Before the measurements were taken, low (at room temperature) and high temperature measurements on water and so-called viscosity standards. First, the dynamic viscosity coefficient of water and reference oil were measured. The following measuring system was used for the following tests, which was later used for measuring liquid steel. The results will be presented as flow curves (dependence between $\dot{\gamma }$—shear rate and η—viscosity coefficient values).

The reference oil used was characterized by low values of dynamic viscosity coefficient, equal to 9 mPas. The authors obtained the viscosity coefficient value about 8.9 mPas in changeable shear rate conditions (between 30 and 90 s^−1^).

The viscosity coefficient values obtained during the measurement of the reference oil are within the limits of the measurement error and confirm that the device has been properly calibrated.

Then, water viscosity measurements were carried out at room temperature, which in the above conditions is a Newtonian liquid with a viscosity of 1 mPas [7]. During the tests, a linear dependence of water viscosity over time was obtained, regardless of changes in shear rate values—the viscosity coefficient value was about 1 mPa s.

Next, at the elevated temperatures, the standard glass (germ. DGG Standardglas I) was used. The chemical composition of the standard glass is presented in a Table 1.

The viscosity value of standard glass at 1400 °C (the highest measured temperature for the referenced glass) should be 14.8 Pas.

The measurement of standard glass was performed in 1400 °C. The obtained results were sufficient. Authors obtained the viscosity value coefficient equal to the 14.8 Pas.

During rheometric measurements of low viscosity liquids, the problem of laminar flow disturbance in the shear gap is a big problem. Then the equations enabling the calculation of the dynamic viscosity coefficient lose their validity [6]. Research on the existence of critical rotational speed for a system with a movable internal cylinder was carried out by Taylor. He first predicted, on the basis of theoretical considerations, and later proved the existence of vortices named after him, which signal the transition of the laminar flow (of the examined medium) into a transient state, and consequently the complex flow-turbulent. In the case of laminar flow, there is no case of layer mixing. Taylor’s vortices testify to the flow linearity disturbance [6]. In order to carry out the measurements correctly, taking into account the fact that there is a critical/limit value of shear rate, for which the flow of liquid steel, characterized by relatively small values of dynamic viscosity, becomes non-laminar—a maximum value of the applied shear rate of 10 s^−1^ was adopted. This value was calculated from the formulas, taking into account the geometry of the concentric cylinder system used and the estimated value of the dynamic viscosity coefficient of the test substance [27].

### 2.2. Analysis of the Chemical Composition of the Samples

Four steel samples were subjected to chemical analysis. The samples for analysis were prepared by cutting and grinding. The analysis was performed using a WAS Foundry-Master optical emission spectrometer. The device is calibrated for analysis of iron alloys. The wavelength measurement range is from 160 to 800 nm. It is controlled by means of a PC, has a vacuum chamber in which a set of CCD photosensitive elements is placed. During the analysis, the spark chamber is rinsed with argon, pre-burning and measurement of the intensity of individual components. The result of the analysis is the percentage chemical composition of the sample.

The following tests were carried out for the iron solutions given in Table 2.

Liquidus temperature values (Table 3) were calculated for each of the above species (Table 2). The calculated temperature values are necessary to determine the measurement scheme, assuming that the steel under test is to be in a completely liquid state.

For each of grades, the Scheil algorithm (representing the non-equilibrium crystallization) was used to calculate the values of their liquidus temperatures [24]:

Thermodynamic databases—CompuTherm LLC—supplied together with the ProCAST software package were used for the calculations.

### 2.3. Experimental Research on Viscosity Measurement

Rheological tests were carried out for three steel grades that differ in their chemical composition (Table 2). The tests were carried out from room temperature up to 1530 °C. From the liquidus temperature (appropriate for each of the iron solutions tested), the measurements were performed every 10 degrees. Tests for each steel grade were carried out twice to verify the results obtained.

Figure 5a shows a steel sample before measurement. The sample has a rectangular shape with a height of 46 mm and a base side equal to 20 mm. Figure 5b shows a steel sample after correctly carrying out measurement. The sample surface did not oxidize. The change in the dimensions of the sample indicates that it completely melted and assumed the diameter of the crucible (30 mm).

Steel tests were conducted in conditions of variable values of rheological parameters ($\dot{\gamma }$—shear rate value in the range of 1–10 s^−1^), the purpose of which was to determine the impact of the above variables on the value of the dynamic viscosity coefficient of liquid steel, and thus to attempt to determine their rheological nature. The tests were carried out from the liquidus temperature value (appropriate for each of the iron solutions tested) up to a temperature of 1530 °C with 10 degree steps. For each value of shear rate, at a given temperature, for a given chemical composition, the measurement was carried out for a minimum of three minutes, with a frequency of data reading every minute. The results were presented in the form of viscosity curves and flow curves.

Figure 6, Figure 7 and Figure 8 present curves for three steel grades. The results are presented for the highest temperature at which the measurements were carried out, i.e., 1530 °C. The results are presented only for this temperature, because the viscosity/flow curves were the same in the case of the tests carried out above the liquidus temperature for each steel grade. Thus, the viscosity results of overheated iron solutions were the same as for these liquids in their liquidus temperature. Due to the fact that the results of the experiments were to be used to model the continuous casting process in the tundish (where temperatures are around 1560 °C), viscosity and flow curves are presented for the maximum temperature achievable in the rheometer—1530 °C.

Viscosity curves are the dependence between $\dot{\gamma }$—shear rate and η—dynamic viscosity coefficient.

Flow curves are the dependence between τ—shear stress and η—dynamic viscosity coefficient [9]. Black lines on the graphs indicate the trend.

In addition, during measurements, it was observed that superheated steels exhibit the same dynamic viscosity coefficient as steels at liquidus temperature (for analyzed steel grades), so it has been assumed that steel at a ladle temperature of 1560 °C will have the same dynamic viscosity coefficient as at 30 °C lower, but still above liquidus temperature of the given alloy.

The course of the curves is similar to the curve of a perfectly viscous fluid. The lowest values of the dynamic viscosity coefficient were obtained for C45 steel—about 0.0255 Pas—while the highest value of the dynamic viscosity coefficient was for 30MnB4 grade—0.0298 Pas. A value of 0.0271 Pas for the dynamic viscosity coefficient was obtained for grade 27MnB4.

Figure 6b, Figure 7b and Figure 8b present flow curve graphs—i.e., the relationship between shear rate and shear stress. This presentation of results is typical for rheological analysis of fluid behavior. In order to verify the influence of the chemical composition on the values of dynamic viscosity coefficient, it is necessary to conduct detailed tests in two-component systems containing iron with an alloying additive, so that it is possible to determine the influence of a given alloying component on the value of dynamic viscosity coefficient. With complex chemical composition, it is difficult to make unequivocal conclusions about the impact of selected elements on the value of the dynamic viscosity coefficient of iron solutions. At this stage, one can only suspect that the increase in carbon content may affect the reduction in the dynamic viscosity coefficient of the iron solution. However, the increase in manganese content seems to cause a slight increase in the value of the dynamic viscosity coefficient of such a solution.

### 2.4. Rheological Models

In the next part of the work, the obtained data were verified using rheological models, namely the Newton model [9] and the author’s empirical MKH1 model [23,27], using the chemical composition of the tested samples for calculation, as well as the rheological parameter (shear stress) and temperature value.

The idea and types of different models which are used to calculate the viscosity of liquids were presented in a previous paper [23,24]. In paper [23], the author described in detail the way to obtain the empirical models dedicated for the calculation of the viscosity of liquid iron solutions.

The purpose of this analysis was to check whether the data obtained through the experiment can be easily calculated using available, selected rheological models. Thus, whether based on data from thermodynamic databases or not, during process modeling, the values are assumed to be similar or significantly different from the values of the dynamic viscosity coefficient obtained through the experiment.

Figure 9, Figure 10 and Figure 11 show viscosity curves for three (selected) of the four iron solutions analyzed. Each of the graphs presented has three viscosity curves: derived from real measurements, calculated from Newton’s model (signed Newton-model), and calculated from the author’s model MKH1 [23,24]. The original MKH1 model was described in detail in [24], and only its mathematical form is presented below:Newton model [9]
(1)$$\eta =\;\frac{\tau }{\dot{\gamma }}$$MKH1 model [23,24]
(2)$$\begin{array}{l} \eta =-\left(0.008626\times {\mathrm{Mn}}^{3}\right)-(0.00372\times \sqrt[{3}]{\mathrm{Ni})}+(0.1036\times \sqrt[{3}]{S\;)}+\\ (0.02244\;\times \sqrt[{3}]{\mathrm{Cu}\;)}-\left(0.00933\times \mathrm{Mo}\right)+\left(0.007434\times C\right)+\\ \left(0.3179\times \mathrm{logT}\right)+\left(0.0187\times \tau \right)-0.129 \end{array}$$

Newton’s model is the most well-known, basic model used to calculate the value of the viscosity coefficient of liquids, both ideal—showing the invariability of the value of the dynamic viscosity coefficient under varying shear stress conditions—and non-Newtonian—showing non-linear dependence between shear rate and shear stress.

In Newton’s model, only values of τ—shear stress and $\dot{\gamma }$—shear rate are taken into account for calculating the η—viscosity value [27].

The MKH1 model is an empirical model based on rheological measurements [23,28]. This model has been developed to calculate viscosity of totally liquid and in the semi-solid state iron solutions.

Multiple linear regression analysis implemented in the STATISTICA version 10.0 and Clementine SPSS version 12.0 software were used for developing the model. To construct the models about 2000 records were used contained the following data: chemical composition—elements concentration in % (C, Mn, Si, P, S, Cu, Cr, Ni, Mo, V); n—rotational speed value; temperature value; $\dot{\gamma }$—shear rate value; τ—shear stress value; η—dynamic viscosity coefficient value. After correlation analysis, only one rheological parameter was chosen as the independent variable. The other independent variables were chemical composition elements and temperature value.

The calculated coefficient of multiple determination for the above equation is R^2^ = 0.842. The Fisher–Snedecor coefficient value is 1259.098; and the critical (read from tables for *n*_1_ = 7, *n*_2_ = 1983) F is 2.01. The hypothesis test conducted with the *p* value confirmed the relationship between the variables analyzed. The standard estimation error is 0.12 [24].

The dynamic viscosity coefficient was considered the output variable, and the other data were entered as input data. Only one rheological parameter—shear stress—was put into the formula.

In the case of the MKH1 model for calculating the viscosity values using the chemical composition of alloy elements (Mn—manganese, Ni—nickel, S—sulphur, Mo—molybdenum, C—carbon), T—temperature, and the rheological parameter is τ—shear stress value and an absolute term.

In the MKH 1 model, the viscosity values of liquid iron solutions are calculated by using its chemical composition, temperature value and rheological parameter (such as shear stress). This point of view is completely different from the viscosity models existing in the literature. Discussions about other types of formulas allowing one to calculate viscosity values (including viscosity of liquid metals) are widely analyzed in another authors’ paper [23,24]. In this paper, the authors would like to show and emphasis that during the modelling of the metal flow (i.e., in the tundish) taking only into account the simple value of the viscosity coefficient of liquid steel or calculating it by using the most well-known model can give improper/faulty values of the viscosity of the liquid iron solution. This, in turn, may cause errors/simplifications of the simulation.

In Figure 9, Figure 10 and Figure 11, the three different viscosity curves presented:With the rhombus-measured values of the viscosity coefficient of investigated liquid iron solutions (Table 2);With the square-calculated values of the viscosity coefficient by using Newton’s model.In this case, the values of shear rate (set during measurement) and obtained shear stress values were put into Newton’s model (1), and the viscosity values were calculated.With the triangular-calculated values of the viscosity coefficient by using empirical MKH1 model.

In this case, the values of obtained shear stress, temperature value, and chemical composition of iron solutions were put into the empirical model MKH1 (2), and the viscosity values were calculated.

The analyses show that the nature of the curve derived from the measurements and the viscosity curve calculated from the MKH1 model are similar. However, the course of the viscosity curve calculated from Newton’s model is different.

The authors would like to show that existing and well-known formulas, such as Newton’s law, might not be sufficient when calculating the viscosity values (during modeling process), and it is necessary to use real data from the experiment or investigate (improve) the empirical model dedicated to calculating the value of the viscosity of liquid high-temperature solutions such as liquid iron alloys.

The nature of the curve derived from the measurements and the viscosity curve calculated from the MKH1 model are similar. However, the course of the viscosity curve calculated from the Newton’s model is completely different.

The model MKH1 rather underestimates the values of the dynamic viscosity coefficient of steel. This is probably due to the fact that the MKH1 model was developed using data obtained for iron solutions with a carbon content of about and above 0.50%. This model should be revised, taking into account a wider range of chemical compositions, including coal. However, graph analysis shows that the Newton model, which is often used in flow modeling programs, incorrectly describes real data obtained during high-temperature rheological studies of iron solutions. Therefore, in order to carry out computer simulations of the real process in which the tested medium is liquid steel, it would be most appropriate to use data from real measurements, and then refine the MKH1 model so that it can be able to calculate the values of the dynamic viscosity coefficient of liquid iron in a wide range of chemical compositions.

In the next part, during numerical simulations of steel flow in the tundish, the constant and experimentally obtained viscosity values of liquid steel were used.

### 2.5. Numerical Simulations of Steel Flow in the Tundish

Tundish model and computational mesh were set with Gambit 2.4 computer program (Ansys, Inc., Canonsburg, PA, USA). Due to the symmetry tundish criteria, the half of the tundish was calculated. Thanks to that, a finer grid could be used with similar computational time. A computational grid was generated for the investigated object (tetra type mesh) with refinement in the area of the wall and submerged entry nozzles (SENs) (mesh was locally refined in areas with high velocity gradients, using the gradient adaption function to obtain a more accurate solution). Mesh quality has been checked with the skew angle criterion. This coefficient, for the computational grid used in the current investigation, is 0.61 and is in the required range. During the adaptation of the computational grid, a dimensionless parameter y+ was taken into account. According to ANSYS Fluent manual, this coefficient should be within 30 to 60, and in the current mesh its value was 54. Thanks to that, the sufficient mesh quality is maintained in this study. The computational domain included 720291 cells (with an average size of 15 mm; in the regions of inlet, the outlet and wall mesh were refined and were about 5–10 mm), this number of elements is sufficient to predict flow structure in the tundish. In a previous investigation done by the author, a similar object was tested with similar initial and boundary conditions, and the results were validated with industrial and water model experiments. Similar mesh size was also proven to be enough for grid independence by other authors [29]. CFD simulations of liquid steel flow through the tundish were carried out with the use of ANSYS Fluent commercial code ver. 16.0. For turbulence modeling, Standard k-ɛ model has been used. Detailed description of the models can be found in ANSYS Fluent Theory Guide [30].

Based on the literature and our own data, a simulation model describing the studied phenomenon was formulated. For non-isothermal calculations, the following boundary conditions were formulated: the bottom and side walls of the ladle meet the conditions of a stationary wall with heat loss of −5 kW/m^2^, and the metal free surface is a wall with zero tangential stresses with heat loss of −12.5 kW/m^2^. Figure 11 presents schematically the boundary conditions used in numerical simulations and mesh used for numerical simulations. At the tundish inlet (shroud), liquid steel flows with a velocity of 1.1 m/s with a turbulence intensity of 5% and the initial temperature of 1823 K.

For the current study, convergence criteria were set to 1e-6 for continuity equation and 1e-3 for other equations. In addition, monitoring points were used in the tundish to test changes in liquid steel velocity between iterations. Monitoring points were set in regions of outlets, inlet and inside the tundish. The solution was found to be converged once the change in liquid steel velocity in monitoring points reached a steady solution and did not exceed 1%.

Numerical calculations were carried out for two variants. In the first variant, the viscosity value used in most calculations, the results of which are widely published (adopted from the literature), in a range about 0.006–0.007 Pas [2,31,32,33], was introduced as the viscosity parameter of liquid steel. The experimentally determined viscosity values (obtained for the steel grades tested in this study) were about four times higher than the literature values. Therefore, in order to perform numerical simulation for the counter-variant, a viscosity value of 0.025 Pas was assumed, which was considered representative for the steel grades tested. Figure 12 shows the distribution of the numerically predicted liquid steel velocity field on transverse tundish planes, successively passing through the shroud and then between each submerged entry nozzle (SEN). Even this cursory qualitative analysis reveals that for both variants, the liquid flow structure in the ladle is very similar, although of course some slight differences can be observed in the flow structure and in the distribution of velocity values.

Figure 13 and Figure 14 show the numerically predicted liquid steel velocity field on longitudinal tundish planes. The first passes through the shroud and is characterized by higher velocity values, and the second passes through outflows (SENs) of the tundish. Similar to the transverse planes, some differences in the distribution of velocity fields can be observed.

The tundish working space can be divided into two characteristic regions. The first of them is located in the area of the shroud (steel inlet from the casting ladle). This tundish area is characterized by a high level of flow turbulence. Liquid steel flows with impetus into the tundish and is directed to the turbulence inhibitor, installed at the tundish bottom, whose main task is to protect the refractory lining of the tundish, reduce the speed of steel and stabilize its flow. This tundish region is characterized by much higher liquid velocity values than the rest of the working area. Dynamic viscosity is negligible in this part of the ladle because eddy viscosity dominates. The remaining part of the ladle is characterized by significantly lower liquid flow velocities (except for small areas at SEN-s neighborhood) and significantly lower turbulence intensity. In this region, differences in the flow structure calculated for individual viscosities can already be observed (Figure 13 and Figure 14). Because treatments such as the flotation of non-metallic inclusions from steel into the slag layer occur mainly in this tundish region, changes in the flow structure, even small and subtle, will affect the prediction of the simulation of non-metallic inclusions’ removal into the slag in this tundish zone. Therefore, it might influence results, especially in death zones, which then might influence, for instance, the prediction of non-metallic inclusion removal. It is important to build up a simulation model as close to reality as possible.

Of course, mesh density and other numerical model parameters have an influence on the computational results, but with the very high mesh density (which with todays computing power is easy to reach) and with the same model settings for both investigated cases, the results can be properly compared with each other.

The quantitative analysis of differences in the forecasted liquid steel speed profiles was carried out by comparing the values obtained for individual lines passing through the tundish working area. The arrangement of the lines is shown in Figure 15 and the results obtained in Figure 16. The lines are located at the following heights: line inlet (shroud) at 0.4 m, line outlet at 0.4 and 0.7 m (ladle is 0.8 m high).

The flow structure and its turbulence changes according to the viscosity value discussed, according to Figure 12, Figure 13 and Figure 14, are confirmed by the results shown in Figure 16. These show velocity magnitude at lines passing through different tundish working zones. Velocity values are given for numerical calculations performed with the standard (base) viscosity value and for viscosity values obtained experimentally. For lines passing through the shroud, and so the region characterized by higher liquid velocity, the values are very close to each other and there are almost no differences. The other situation can be seen in other regions of the tundish, where liquid steel velocities are lower. Velocity lines obtained for lines passing through the submerged entry nozzles, for both highs (0.4 and 0.7), slightly differ from each other. Again, in regions closer to the liquid steel inlet (shroud), the values are very similar, but in the more distant zones, the velocity profile changes and the differences even reach about 20%. This was observed at both fluid levels and can be very important in the liquid surface area, were non-metallic inclusions can react with covering slag.

Both qualitative and quantitative analysis showed differences in the flow structure of liquid steel in the tundish, in particular in the predicted values and the velocity profile distribution. However, these differences are not significant, and in most of the analyzed working areas of the tundish, they are negligible. On the other hand, one should not uncritically take values from the literature data, especially if the case deviates from others. Although there are other effects that have more influence on the simulation results than the presented difference in viscosity, the current investigation is focused on basic research and the purpose is show that the choice of correct viscosity values might influence the results, especially in death zones, which then might influence, for instance, the prediction of non-metallic inclusion removal or when one investigates the slag re-entrainment process into the liquid steel. It is important to build a simulation model as close to reality as possible.

To sum up, if there is such a possibility, the value of steel viscosity should be determined by means of experimental tests, while in the absence of such a possibility by entering a value from publicly available results of experimental tests in this area, it should be ensured that they are as close as possible to the analyzed case—then, when numerically forecasting the flow found in the ladle, one will not make a significant mistake. However, this further emphasizes the importance of conducting experimental research to determine the viscosity of various steel grades.

## 3. Conclusions

Based on the results of the experimental measurements and numerical simulations carried out, the following conclusions were drawn:A higher amount of carbon in the chemical composition of steel causes a lower liquidus temperature. At the investigated temperature, this iron solution probably does not have solid particles which could affect its rheological behavior. Steel C45, with the higher content of carbon, has the lowest value of the viscosity coefficient.With complex chemical composition, it is difficult to make unequivocal conclusions about the impact of selected elements on the value of the dynamic viscosity coefficient of iron solutionsThe character of viscosity and flow curves, obtained from rheological measurements, is quite similar to the course of a perfectly viscous fluid curve.The nature of the curve derived from the measurements and that of the viscosity curve calculated from the MKH1 model are similar.The character of the viscosity curve calculated from Newton’s model is completely different.Existing and well-known formulas, such as Newton’s law, might not be sufficient when calculating the viscosity values—during the modeling process—of liquid iron solutions.The author’s rheological model still needs to be refined and extended to a wide range of alloying elements.Based on numerical predictions, qualitative and quantitative analysis showed differences in the flow structure of liquid steel in the tundish.Differences received for calculations with different steel viscosities are not significant, and in most of the analyzed working areas of the tundish, they are negligible.Nevertheless, investigations are focused on basic research and it was shown that the choice of correct viscosity values might influence the results, especially in death zones, which then might influence, for instance, the prediction of non-metallic inclusion removal or dragging slag into the metal.

## Figures and Tables

**Figure 1 materials-13-05025-f001:**
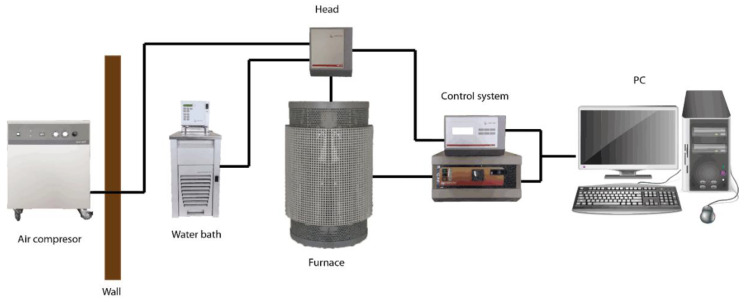
Diagram of FRS1600 high temperature rheometer.

**Figure 2 materials-13-05025-f002:**
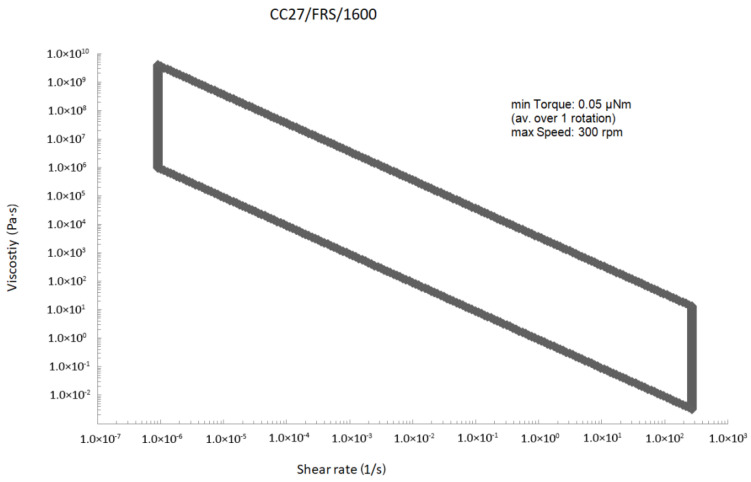
An envelope of the rheometer measuring range using a 26.6 mm spindle.

**Figure 3 materials-13-05025-f003:**
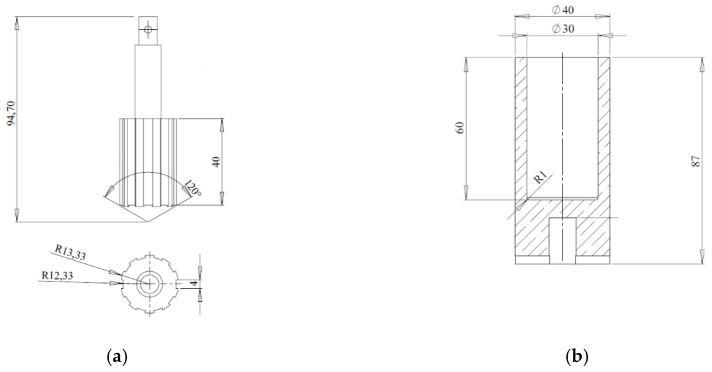
Measuring tools used for research, (**a**) bob; (**b**) cup.

**Figure 4 materials-13-05025-f004:**
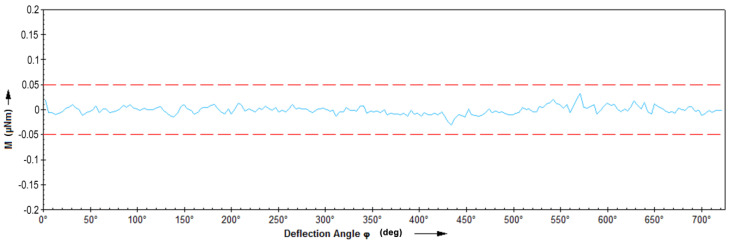
Air test chart.

**Figure 5 materials-13-05025-f005:**
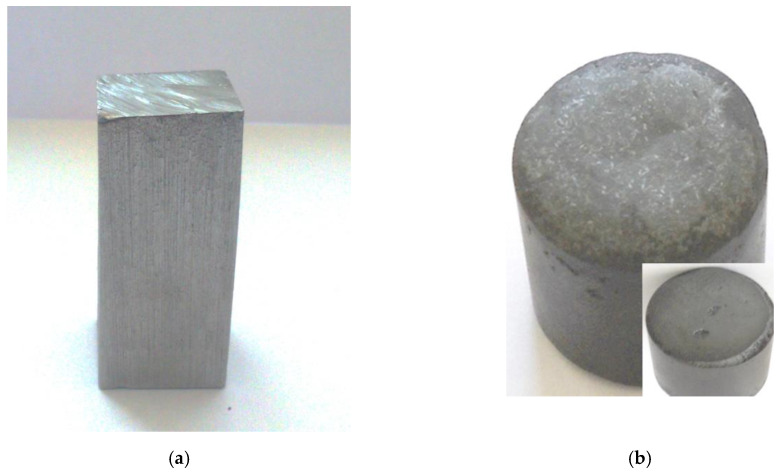
(**a**) Steel sample before measurement; (**b**) steel sample after measurement.

**Figure 6 materials-13-05025-f006:**
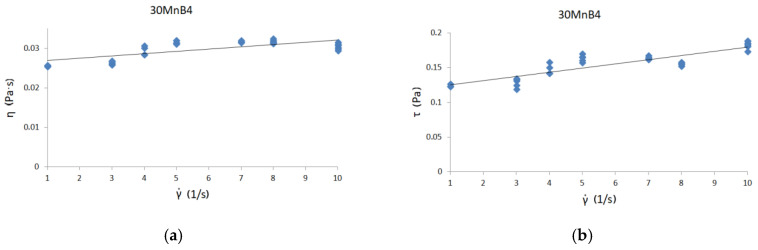
(**a**) Viscosity curve for 30MnB4 steel at 1530 °C; (**b**) flow curve for 30MnB4 steel at 1530 °C.

**Figure 7 materials-13-05025-f007:**
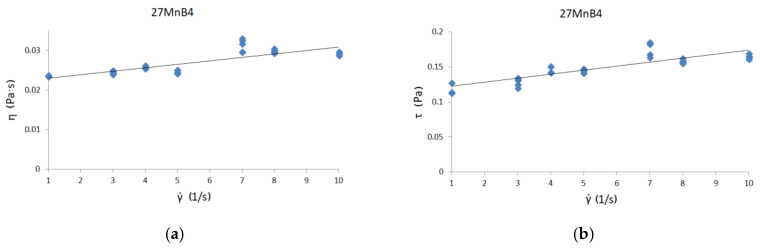
(**a**) Viscosity curve for 27MnB4 steel at 1530 °C; (**b**) flow curve for 27MnB4 steel at 1530 °C.

**Figure 8 materials-13-05025-f008:**
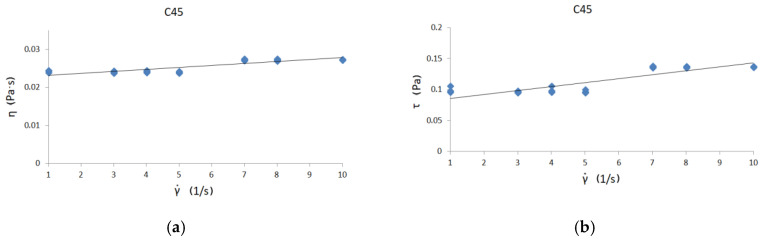
(**a**) Viscosity curve for C45 steel at 1530 °C; (**b**) flow curve for C45 steel at 1530 °C.

**Figure 9 materials-13-05025-f009:**
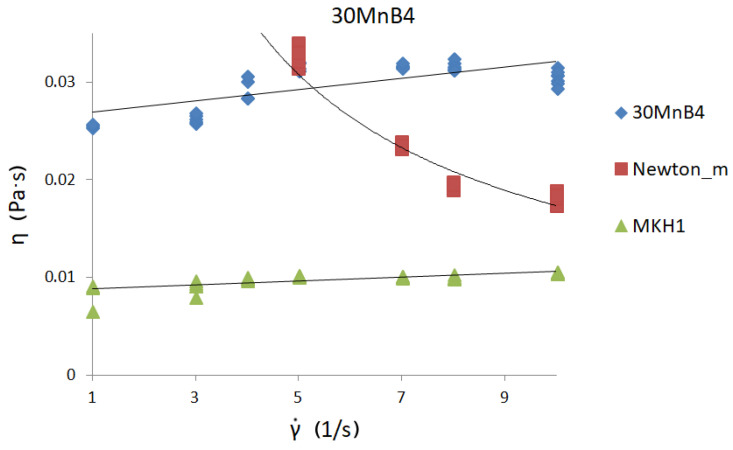
Viscosity curve for 30MnB4 steel at 1530 °C.

**Figure 10 materials-13-05025-f010:**
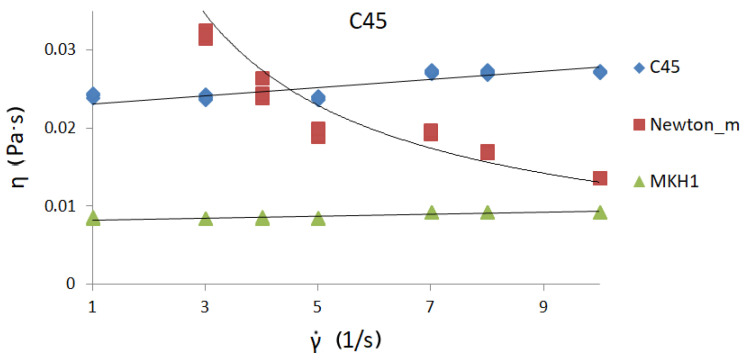
Viscosity curve for C45 steel at 1530 °C.

**Figure 11 materials-13-05025-f011:**
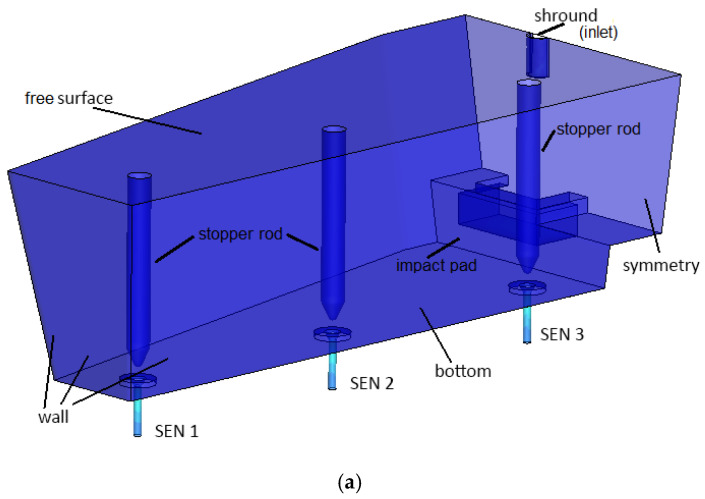
Investigated tundish with (**a**) boundary conditions and (**b**) mesh used for the simulations.

**Figure 12 materials-13-05025-f012:**
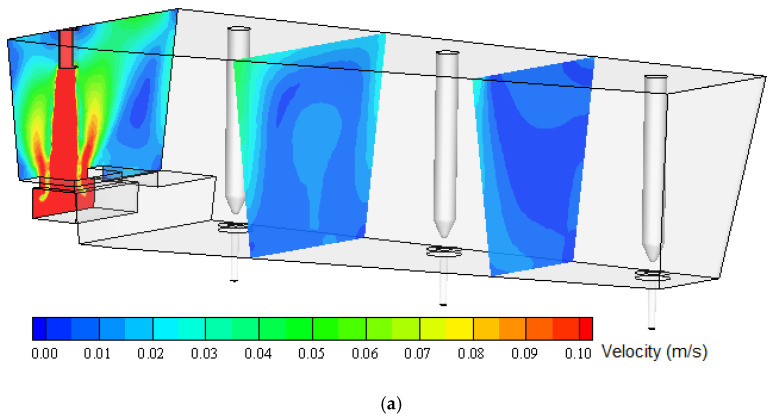
Numerically predicted velocity field of liquid steel on transverse tundish planes: (**a**) base viscosity, (**b**) experimentally obtained viscosity.

**Figure 13 materials-13-05025-f013:**
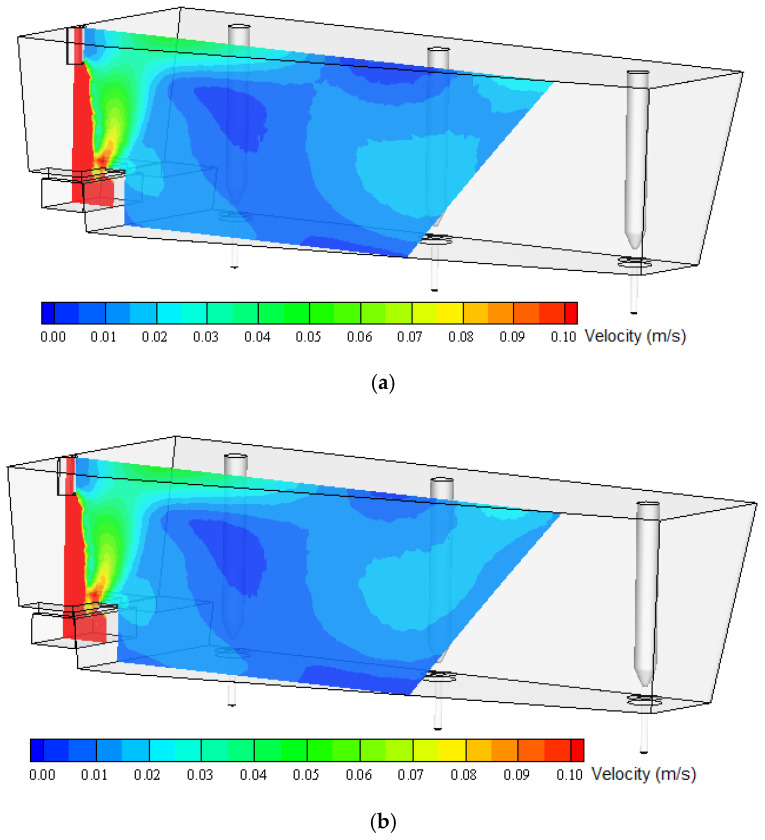
Numerically predicted velocity field of liquid steel in the longitudinal tundish plane passing through the shroud: (**a**) base viscosity, (**b**) experimentally obtained viscosity.

**Figure 14 materials-13-05025-f014:**
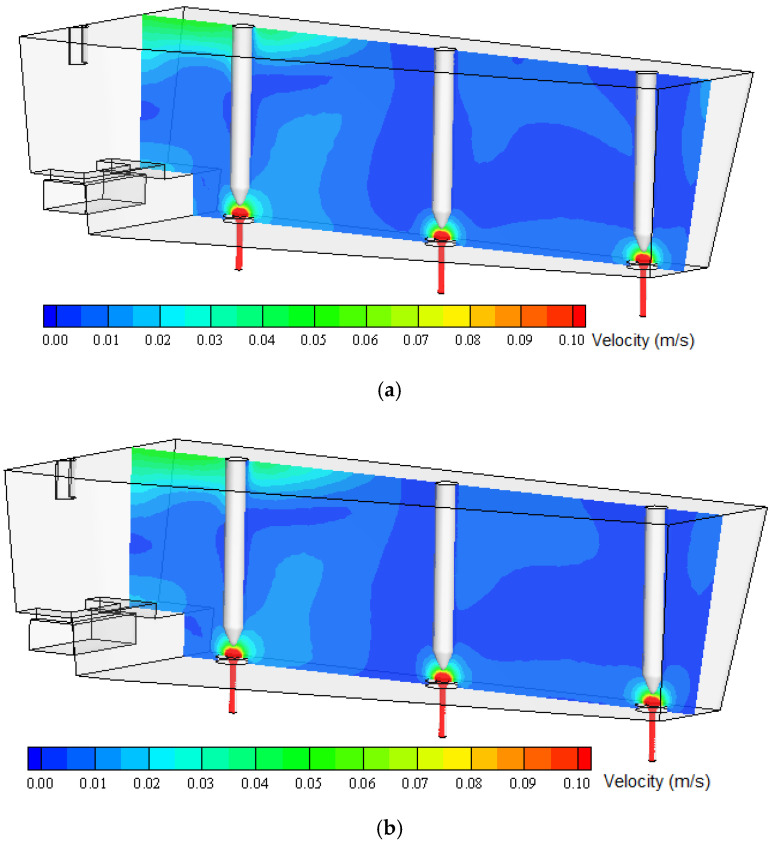
Numerically predicted velocity field of liquid steel in the longitudinal tundish plane passing through submerged entry nozzles (SENs): (**a**) base viscosity, (**b**) experimentally obtained viscosity.

**Figure 15 materials-13-05025-f015:**
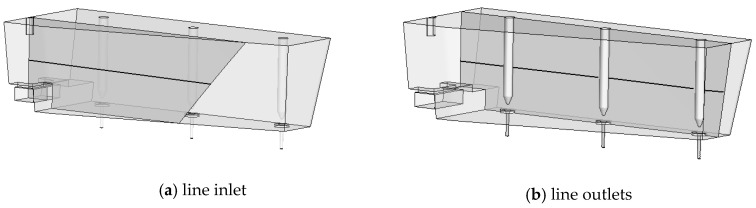
Location of measuring lines: (**a**) passing through the shroud, (**b**) passing through the SENs.

**Figure 16 materials-13-05025-f016:**
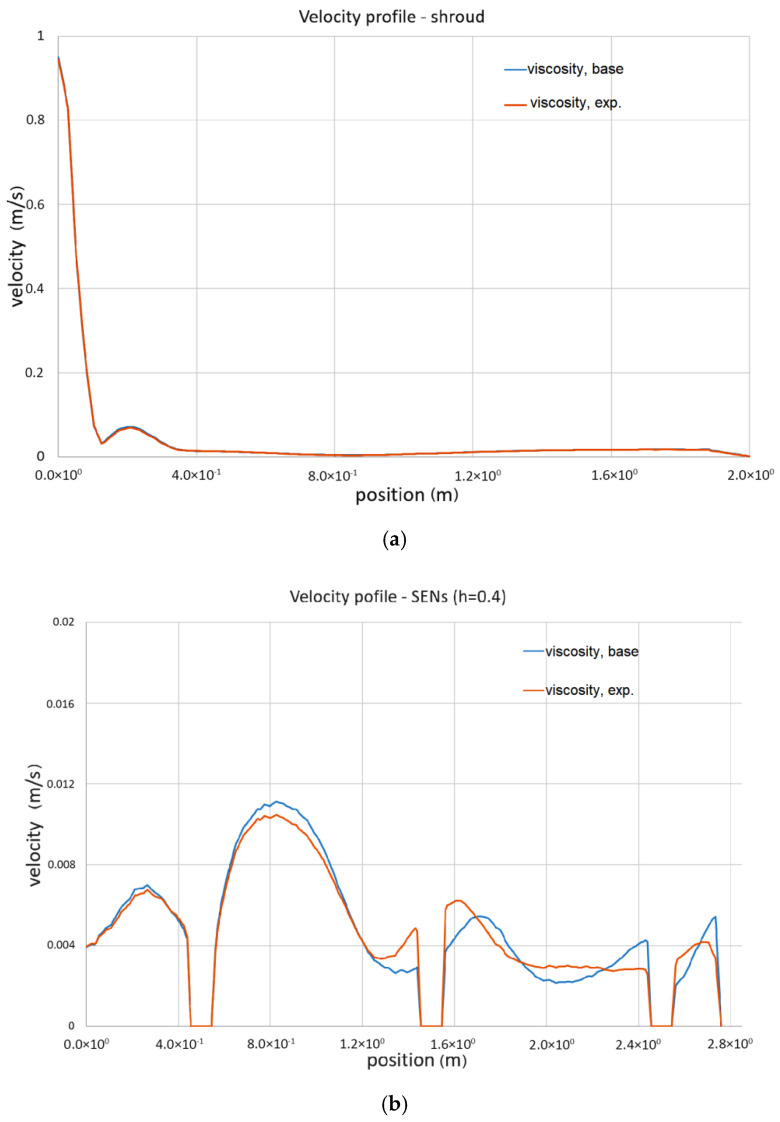
Liquid steel velocity value on lines passing through shroud and outflows (SENs); (**a**) shroud; (**b**) SENs, h = 0.4; (**c**) SENs, h = 0.7.

**Table 1 materials-13-05025-t001:** Chemical composition of standard glass.

Chemical Composition	SiO_2_	Al_2_O_3_	Fe_2_O_3_	TiO_2_	SO_3_	CaO	MgO	Na_2_O	K_2_O
%	71.72	1.23	0.191	0.137	0.436	6.73	4.18	14.95	0.338

**Table 2 materials-13-05025-t002:** Chemical composition of investigated alloys.

Steel Grade	C	Mn	Si	Cr	Ni	Mo	B
30MnB4	0.289	0.9	0.11	0.19	0.56	0.01	0.002
27MnB4	0.293	0.98	0.25	0.21	0.07	0.008	0.002
C45	0.41	0.68	0.24	0.08	0.07	0.01	

**Table 3 materials-13-05025-t003:** Liquidus temperature values for the analyzed steel grades [°C].

Steel Grade	Liquidus Temperature (Scheil Algorithm)
30MnB4	1511
27MnB4	1509
C45	1503

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
