# Peer review of "Investigations of Liquid Steel Viscosity and Its Impact as the Initial Parameter on Modeling of the Steel Flow through the Tundish"

_materials, 2020, doi:10.3390/ma13215025_

Round 1

Reviewer 1 Report

The paper has been improved.

But,

Again, the MKH1 model is poorly explained. One can go to refs 23 and 27, but just a few lines explaining it better must be added. What is Ni, S etc? Concentrations?

in the text you say:

To construct the models about 2000 records were used contained the following data: chemical composition (C, Mn, Si, P, S, Cu, Cr, Ni, Mo, V)

- so concentrations I guess - in %? or from 0 to 1 being 1 100%?

rotational speed value - where in the formula ??

temperature value;

T

– note here, lgT is logT (base 10) or lnT ? (just to clarify)

shear rate value;

γ̇ – where is in the formula?

shear stress value;

τ – right?

dynamic viscosity coefficient value.

η I assume

and − 0,129 ?? meaning??

and finally,

> Graphs made with Excel

There are better programs to do them. I do not see a reason to use excel.

Also about graphs,

Maybe graphs 6a, 7a, 8a and 9a could be just one, as well as 6b, 7b, 8b and 9b, could be presented in only one?

For comparison shake could be useful.

Finally. Are lines in all these graphs linear fits or guides to the eye?

Author Response

Thank You for revising of our manuscript. Reply to the Review Report in attachment.

Reviewer 2 Report

The article still needs deeper modifications and reworking, especially the first experimental part. Not all questions were answered and implemented in the article.

I also don't see the logical continuity between the first (experimental) part and the second (numerical simulation) part of the article.

The first part states that the dynamic viscosity does not depend on the shear rate or on the superheating of the steel above the liquid, but in the second part it is shown that the effect of viscosity on the flow is negligible. The connection between the two parts is very limited.

I recommend a thorough reworking and expanding of each part and publishing them in two separate articles.

row 218, 528: Above the liquidus temperature, all mass should be liquid, without solid particles (if the steel is free of impurities), contrary to what you say on line 528. Are liquidus temperatures correct? If three algorithms have been used to calculate the liquidus temperature, three results and their differences should be given. There are also a number of empirical equations in literature that can also be compared.

row. 249: „…it was verified that superheated steels exhibit the same dynamic viscosity coefficient as steels at liquidus temperature, so it can be assumed that steel at a ladle temperature of 1560 °C will have the same dynamic viscosity coefficient as at 30 °C lower“.

In scientific work, this statement should be substantiated by experiment data or by reference to the literature.

row. 279 and Fig. 6-9: You wrote you verified the measured data. But I still think that your measurement results are doubtful. For known materials the shear stress is not independent of the shear rate, unlike your measurements. In case of Newtonian fluids this dependence is linearly increasing etc. What is more, if you claim (in the row 279), that liquid steel is a Newtonian fluid, both the shear stress and the apparent dynamic viscosity cannot be independent of the shear rate at the same time (e.g. Fig. 9a and 9b).

row. 238: „The tests were carried out from a temperature of 1530 °C in 10 degree steps, up to the liquidus temperature value appropriate for each of the iron solutions tested.“

This sentence is wrong, liquidus temperatures are lower than 1530 °C.

row. 316: If you claim that liquid steel is a Newtonian fluid and that viscosity does not depend on overheating above the liquid, why are the shear stress and temperature in the regression model among independent variables? Btw, the empirical model should be supplemented by an analysis of the statistical significance of the coefficients for individual independent variables.

Author Response

(The authors gave the same response as above.)

Reviewer 3 Report

The author measured the viscosity of liquid steel, and studied the effect of liquid steel viscosity on the liquid steel velocity.  The experimental data and the related numerical result is helpful to the metallurgists. But the paper has the following questions.
(1) Line 364 "Due to the similarity criteria, " should be replaced by "Due to the symmetry,".
(2) Grid system is the key factor to affect the numerical result. Especially,  the tundish in Fig 13 has a complex structure. Please give the grid system.
(3) Please give the grid independence test.
(4) Please give the convergence criterion.
(5) Line 403 "tundih" should be "tundish".
(6) Standard k-epsilon model is applied in the numerical simulation. In k-epsilon model, effective viscosity ueff=u0+ut. Usually, u0<<ut. This means the molecular viscosity is far less than the turbulent viscosity.  In other words, the effect of the molecular viscosity on the numerical result can be ignored. 

Author Response

(The authors gave the same response as above.)

Reviewer 4 Report

This paper presents research carried out to experimentally determine the dynamic viscosity of different iron solutions, and numerical simulation has been established to validate the experimental results. Also, the flow structure of liquid steel in the tundish under different viscosity has been compared qualitatively and quantitatively. The theme and the content of this manuscript seem to be interesting. However, there still exists some points should be revised, all of these are commented on the manuscript PDF file and listed below:

  1. There lacks of Ref.9 in Line 64. Please make a revision.
  2. In Line 115-137, you introduced the device of FRS1600 high temperature rheometer, but this device is used to measure the viscosity. Please give an introduction about the mechanism how the viscosity measured on this rheometer.
  3. 2 shows the relationship between shear rate and viscosity. However, I can not understand the meaning of this figure, please give an introduction about this figure.
  4. In line 188, the sentence ‘Author obtained the viscosity value coefficient equation to the 14.8 Pas’. Does this value have been published in author’s previous paper, If yes, please give the relevant reference. Others, please give a figure to validate your data.
  5. Fig 5(a)and (b) show the steel sample before and after measurement, what does this figure mean, please give an explanation.
  6. Figures 6-9 present curves for different grades.
    a) What does the abscissa γ indicate, is it the shear rate? Please give an explanation
    b) what does the vertical coordinate τ indicate, please give an introduction on the relationship between τ and flow curve.
    c) Does the black line in figure obtain by fitting? Please explain in the text.
  7. Figure 13 is not clear, please add more explanation, for example, the free surface, the stopper and tundish inlet, etc.
  8. Please label the Velocity headline in legend of Fig. 14-16.

Author Response

(The authors gave the same response as above.)

Round 2

Reviewer 2 Report

The article remained essentially almost in its original form. In many cases, the authors only explain their views, but the significant comments of the reviewer were not accepted. I repeat and supplement what was said in the previous review.

The second part of the article on numerical simulations has a certain information value as a sensitivity analysis of the effect of viscosity on the flow in the tundish. I recommend expanding and publishing the second part separately.

I cannot recommend the first part for publication.

The measured results of viscosity and shear stress seem illogical and physically impossible. E.g. viscosity and shear stress cannot be independent of shear rate at the same time.

Authors have supplemented the empirical regression model with statistical tests which assess the model as a whole (by R2, F-test, p-value etc.). But in my previous comment, I recomended testing the influence on viscosity of individual independent variables that are in the regression model. It should explain why there is temperature in the model if you claim that it does not affect the viscosity? Also shear stress as an independent variable in the model makes no sense as it is tied together with viscosity over shear rate which is given by speed of the apparatus. Overall, the text is inconsistent.

Author Response

Thank you for rigorous study and opinion of the paper. Please, find our answers in attached file. 

Reviewer 3 Report

The author gave the equation between the viscosity and chemical conent in the molten steel, and showed the effect of the visocity of molten steel on the fluid flow in the tundish. Such research can receive the metallurgists' interest. But there are some problem. 

(1) Please check the grid independence.

(2) Please give the mesh system.

(3) Please give the convergence criterion.

Author Response

Thank you for rigorous study of our article. Our answers are presented in attached file.

This manuscript is a resubmission of an earlier submission. The following is a list of the peer review reports and author responses from that submission.

Round 1

Reviewer 1 Report

I do not see any change in your manuscript. The CFD section is irrelevant to the content of the paper.

Reviewer 2 Report

I see the paper has been improved

but I still have two main comments

First is about the figures that can certainly be improved, all of them.

>The quality of the figures has been improved. Authors also tried to unify the font and styles

of the figures, however sometimes it was impossible because some of them are prepared by the commercial software (i.e. Rheoplus) and all styles and fonts are already defined in it.

I do not know about that machine, but I guess the machine provides the experimental data in text files as well (?). Then, what’s the problem to make all plots with some software like Origin?

Second:

In general, the model, that authors claim is well explained in the references, should be at least, briefly explained in the text.

Equation 2 has to be written using some parenthesis. Otherwise is not clear.

3

check english

Mn is Manganese, no manganic

Reviewer 3 Report

General remarks:

The article consists of two parts. The first part presents the results of viscosity measurements and calculated values. Measurement results are questionable. If the viscosity does not depend (or depends weakly) on the velocity rate (e.g. Figure 9a), then the shear stress cannot be independent on the velocity rate at the same time (unlike e.g. Figure 9b). This logically follows the strange hyperbolic dependences of the viscosity on the velocity rate (Fig. 10-12).

The regression formula (model MKH1) is applied to the values of the input parameters outside the validity area of the formula, except for the steel C60. Results are graphically displayed and compared with measurements. Although the formula is applied outside its validity range, it gives results closer to commonly used literature data than measured values of viscosity. Is the measurement of the molten steel viscosity just above the liquidus temperature in the instrument correct? Has it been verified on another device? Is the calculated liquidus temperature correct?

In the second part, an interesting sensitivity analysis of the influence of viscosity on the character of the flow in the tundish is performed using a numerical model of the flow in the tundish. Up to 4-fold difference in viscosity has minimal effect on flow characteristics and velocities.

Notes on the text:

As lines of the text are not numbered, reviewing and writing notes is a little bit uncomfortable.

According to the editor’s instructions, the units on the graph axes should be in parentheses or not?

page 5: The description of the process of the rheometer calibration is detailed, but on the other hand, it is not stated what is the accuracy of temperature measurement and its uniformity in the retort. How is torque and speed measured?

page 6: It was not necessary to describe the well known principle of the spectrometer. On the other hand, it is not stated how the liquidus temperatures were calculated. It is known that there are many mathematical models that differ significantly. Is overheating of steel samples above the liquidus temperature real and sufficient?

The tests were carried out from the room temperature up to 1530 °C in 10 degrees steps, up to the liquidus temperature value“. The sentence does not make sense. From the room temperature? What is correct, up to 1530 °C or up to the liquidus temperature? Liquidus temperatures of the samples are lower than 1530°C.

page 7: Similar sentence is repeated: “The tests were carried out from a temperature of 1530 °C in 10 degree steps, up to the liquidus temperature…“ Didn't the author mean "above the liquidus temperature" instead of „up to the liquidus temperature“? The same was written at page 6. But measurements were carried out at a single temperature 1530°C.

… the measurement was carried out for a minimum of three minutes, with a frequency of data reading every minute“. It means that only 3 measured values were obtained in 3 minutes?

„…it was verified that superheated steels exhibit the same dynamic viscosity coefficient as steels at liquidus temperature“. There is no evidence in the text (no literature citation or experiment) that viscosity of steels above liquidus temperature does not depend on temperature.

“...completely liquid iron solution is a Newtonian fluid…“. Viscosity of a Newtonian fluid does not depend on velocity rate, which does not agree with the measured results, e.g. case of the steel 27MnB4.

Page 8. Similarly: “It can therefore be concluded that the completely liquid iron solution is a Newtonian fluid“. Similarly to the previous comment, the claim that the liquid steel is a Newtonian fluid is too strong according to the performed measurements. How viscosity and shear stress can be constant at the same time (see Fig. 9a and 9b)? Shear stress should change almost 10 times in the measured range of velocity rate.

The title of Figure 9b contains wrong steel grade C45.

Page 9: In the equation (1) the symbol for shear rate is y with a dot, but in Figures 6-9 and 10-12 the dot is missing. Shear rate equals to velocity rate?

Newton's model is the most well-known, basic model used to calculate the value of the viscosity coefficient of an ideal liquid…“ In fact, the equation (1) is valid for all liquids, including non-Newtonian. Only in case of Newtonian liquids the fraction is constant.

Pages 10-11: The strange hyperbolic course of Newton_m curve and extreme values (Fig. 10-12) are caused by almost constant shear stress and viscosity (independent on velocity rate). Measured values should be verified.

Wrong unit on the vertical axis in the Figure 12.

If the model MKH1 was derived from data for steel with carbon content equal and higher than 0,5 %, it is pointless to use it outside this validity interval. The model should be calibrated according to new measurements.